# CONFLO: Conformal Prediction with Conditional Coverage via Normalizing Flow

## Abstract

Beyond point predictions, conformal prediction provides prediction sets that enjoy finite-sample probability coverage guarantees, but only at the population level. In practice, prediction sets can under- or over-cover in subpopulations, limiting their usability for individual predictions. To address this issue, we propose CONFLO, a conformal prediction framework that integrates conditional normalizing flows (CNF) with a novel form of regularization. The anchoring idea is to transform raw nonconformity scores through a feature-dependent bijection into new scores that are (nearly) independent of the inputs. Since independence cannot be perfectly achieved in practice, we add a quantile-aligning penalty to the loss function as an additional tactic to enforce common conditional coverage across user-specified groups. Experiments on diverse datasets demonstrate that CONFLO improves conditional coverage across subpopulations and sizes of prediction sets compared to baseline methods and competitors like APS and RAPS.

## 1 Introduction

Conformal prediction is a widely adopted method in supervised learning for constructing prediction sets with distribution-free, finite-sample coverage guarantees. The method is especially appealing in high-stakes applications where robustness and reliability of outcome prediction are essential. However, conformal prediction guarantees only *marginal* coverage. That is, the prediction set contains the true outcome with the desired probability when averaged over the population of test points. In many applications, we would prefer a stronger guarantee called *conditional* coverage, meaning that the coverage probability is achieved in each subpopulation of interest, say, when conditioning on a particular input value. Unfortunately, it has been shown theoretically that exact conditional coverage is unattainable in a distribution-free and finite-sample setting (Vovk, 2012b; Barber et al., 2019).

This finding has motivated a series of approximations of the conditional coverage condition. Andrews & Shi (2013) showed that conditional moment restrictions can be reformulated as unconditional ones through carefully chosen instrument functions, an idea that later inspired functional relaxations of conditional coverage. Vovk (2012a) formalized several notions of conditional validity and established the impossibility of exact object-conditional guarantees, highlighting the need for approximate alternatives. Building on these, Hebert-Johnson et al. (2018) introduced multicalibration, requiring calibration across all computationally identifiable subgroups, thereby strengthening conditional reliability. More recently, Gibbs et al. (2024) unified these directions by casting conditional coverage as a family of moment conditions, yielding exact guarantees for each subgroup if there is a finite number of them, and approximate guarantees in more general settings.

Focusing on achieving good conditional coverage in practice, Romano et al. (2020) proposed Adaptive Prediction Sets (APS), which replace fixed score thresholds with adaptive stopping rules along the ordered list of class probabilities, thereby reducing excessive set sizes in easy cases while still maintaining coverage. Extending this idea, Angelopoulos et al. (2022) developed Regularized Adaptive Prediction Sets (RAPS), which combine APS with penalties on large or redundant sets and optional inclusion of low-probability "buffer" classes. RAPS achieves tighter and more stable conditional coverage, especially in multi-class and high-dimensional problems such as image classification.

For another work that is highly related to this paper, Colombo (2024) introduced a Normalizing Flow (NF) based framework for conformal regression when the outcome variable is continuous and

one-dimensional. Their method learns a feature-dependent transformation of the residuals, which serve as the nonconformity scores, by fitting an NF that transforms the joint distribution of the residuals and the features into a representation that has near independence between a new score and the features. This produces prediction intervals that adapt to local heteroscedasticity.

Our approach, CONFLO, is inspired by the same principle of using flow transformations to nearly eliminate the dependence of non-conformity scores on features, but differs in two key aspects: (i) CONFLO is applicable to outcomes of any type, including one and multi-dimensional numerical, and categorical ones. This is achieved as follows: instead of applying an NF to the joint distribution of residuals and the input, CONFLO applies a CNF to any user-chosen nonconformity score conditional on the input; (ii) Rather than depending entirely on the transformation for near-independence, CONFLO pushes for groupwise quantile alignment by adding a penalty term to the training objective of the CNF. After all, CONFLO uses all calibration data to form a single threshold, yielding prediction sets with near-nominal conditional coverage.

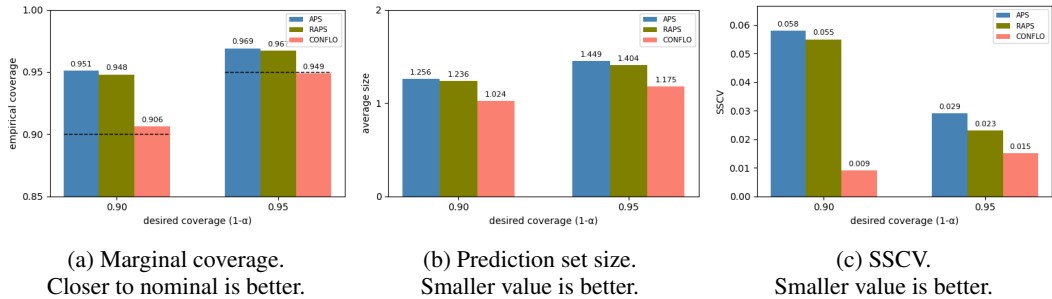

| (a) Marginal coverage. | (b) Prediction set size. | (c) SSCV. |
| Closer to nominal is better. | Smaller value is better. | Smaller value is better. |

Figure 1: Empirical coverage, average set size and SSCV across three methods designed to provide conditional coverage: APS, RAPS, and CONFLO. All methods use a transformer encoder (DistilBERT) with a linear head. Results shown are from 20 random splits of AG News dataset into training, calibration and testing

.

For an example, Figure 1 summarizes the performance of APS, RAPS, and CONFLO when applied to an `AG News` dataset (Li, 2024). The dataset contained the title and description of 20,000 news pieces represented as text features ($\mathbf{x}$), with class labels ($y$) taking values in `World`, `Sports`, `Business`, or `Sci/Tech`. Panel (a) demonstrates near-target coverage across methods, with APS and RAPS overshooting and CONFLO aligning closely. Panel (b) highlights CONFLO 's efficiency, producing smaller sets while maintaining coverage. Panel (c) displays a metric called SSCV introduced in Angelopoulos et al. (2022) and defined in equation 7, which is the maximum deviation from the nominal coverage level of any subgroup by prediction set size. Overall, we can see significant improvement in marginal coverage, prediction size and conditional coverage by the propsed CONFLO. More empirical examples that shows similar results are provided in Section 4.

## 2 CONFLO: Conformal Prediction with Conditional Coverage via Normalizing Flow

### 2.1 Problem setup

Let $\mathcal{X}$ denote the input (feature) space and $\mathcal{Y}$ the outcome space. $\mathcal{D} = \{(\mathbf{x}_i, y_i)\}_{i=1}^{n}$ is an i.i.d random sample from some joint population distribution $P$ over $\mathcal{X} \times \mathcal{Y}$. The objective is to construct a *prediction set*, $\widehat{C}_\alpha(\mathbf{x}) \subseteq \mathcal{Y}$, for a new subject drawn from the population with feature $\mathbf{x}$, that provides a set of plausible values of its corresponding outcome, $y$, beyond a single point prediction. The standard requirement for a prediction set is *marginal probability coverage*:

$$\mathbb{P}\{Y \in \widehat{C}_\alpha(\mathbf{X})\} \geq 1 - \alpha. \tag{1}$$

A stronger goal is to also achieve *(approximate) conditional coverage*:

$$\mathbb{P}\{Y \in \widehat{C}_\alpha(\mathbf{X}) \mid \mathbf{X} = \mathbf{x}\} \approx 1 - \alpha \quad \forall \, \mathbf{x} \in \mathcal{X}.$$

To obtain marginal coverage, a most common approach is the split conformal approach (Papadopoulos et al., 2002; Vovk et al., 2005; Lei et al., 2015). Briefly, the approach first split $\mathcal{D}$ into a training set, $\mathcal{D}_{\text{train}}$, used to fit a predictive model $\hat{f}$, and a calibration set, $\mathcal{D}_{\text{cal}}$, used to quantify the uncertainty of $\hat{f}$ via a key term called *non-conformity scores*, which we denote by $a(\hat{f}, \mathbf{x}, y)$. Taking the classification problem with $\mathcal{Y} = \{1, \ldots, K\}$ for example, the classifier can be represented as $\hat{f}(\mathbf{x}) = (\hat{f}_1(\mathbf{x}), \cdots, \hat{f}_K(\mathbf{x}))$ where $\hat{f}_k(\mathbf{x})$ denotes the predicted probability of the subject belonging to the class $k$. A commonly used nonconformity score is then given by

$$a(\hat{f}, \mathbf{x}, y) = -\log\left(\hat{f}_y(\mathbf{x})\right). \tag{2}$$

Let $Q_{1-\alpha}$ be the $\left(\frac{n+1}{n}(1-\alpha)\right)$th empirical quantile[1] of $\{A_i = a(\hat{f}, \mathbf{x}_i, y_i), i \in \mathcal{D}_{\text{cal}}\}$. Then for a new input $\mathbf{x}$, the prediction set,

$$\widehat{C}_\alpha(x) = \{y \in \mathcal{Y} : a(\hat{f}, \mathbf{x}, y) \leq Q_{1-\alpha}\} = \{y \in \mathcal{Y} : -\log\left(\hat{f}_y(\mathbf{x})\right) \leq Q_{1-\alpha}\},$$

satisfies the marginal coverage property as defined in equation 1. In words, the prediction set consists of all classes whose scores fall below the "global" quantile threshold, $Q_{1-\alpha}$.

Note that the distribution of $A \mid \mathbf{X} = \mathbf{x}$ typically depends on $\mathbf{x}$ and can vary substantially across different feature values. Hence the quantiles $Q_{(1-\alpha, \mathbf{x})}$ of subpopulations, $A|\mathbf{X} = \mathbf{x}$, will deviate from the global quantile (the population version of $Q_{1-\alpha}$). Consequently, using the same threshold $Q_{1-\alpha}$ to form prediction sets for $\mathbf{x}$ across regions of $\mathcal{X}$ will lead to under-coverage in some subpopulations and over-coverage in others. To address this issue, prior work has relied on coarse, subjective partitioning of the data by $\mathbf{x}$ or $y$ values, fitting separate quantiles within each group. In contrast, our method CONFLO first transforms the score $A$ into a new score $B$ that is nearly independent of $\mathbf{X}$, so that the new score across different $\mathbf{x}$ share the same conditional distribution hence common global quantile. This yields two key benefits: greater efficiency by using all data to estimate one common quantile at any chosen level $(1-\alpha)$ in calibration, and prediction sets that (nearly) achieve conditional coverage for new subjects.

Below, we describe how this transformation can be trained to achieve near independence in a CNF framework and a novel penalty function.

## 2.2 Transformations towards Independence

Bogachev et al. (2005) has shown that there always exists a strictly monotone map $t^\star(a, \mathbf{x})$ such that the transformed scores
$$B = t^\star(A, \mathbf{X})$$
satisfy the distributional invariance,

$$F_{B|\mathbf{X}=\mathbf{x}}(b \mid \mathbf{x}) = F_B(b), \qquad \forall \mathbf{x} \in \mathcal{X},\ b \in \mathbb{R}.$$

In this ideal case, marginal and conditional distributions coincide, and exact conditional coverage follows. Since the ideal transformation $t^\star$ is unknown, we resort to a parametric family $\{t_\theta(a, \mathbf{x}) : \theta \in \Theta\}$ and assume that $t^\star$ can be well-approximated within it. Furthermore, rather than conditioning on the full high-dimensional input $\mathbf{x}$, one may use a lower-dimensional representation $h(\mathbf{x})$ learned during training. Such dimensionality reduction can stabilize the training of $t$ and yield closer approximations to conditional coverage across levels of $h(\mathbf{x})$. To simplify notation, we use $\mathbf{x}$ from now on to denote either the full vector of features or a reduced representation, omitting explicit reference to $h(\mathbf{x})$.

### 2.2.1 Conditional Normalizing Flows

We approximate the ideal transformation $t^\star$ via CNF (Chen et al., 2019; Kingma & Dhariwal, 2018; Winkler et al., 2019), which is a family of invertible and differentiable mappings that can transform a collection of random vectors with complex conditional distributions to user-specified base distributions. Each CNF transformation is determined by a vector of parameters $\theta$, and the transformed

---

[1]Here, the $p-$th empirical quantile of a set is defined to be the $\lceil p \rceil$th smallest order statistic of the set, where $\lceil \cdot \rceil$ is the ceiling function.

scores are $b = t_\theta(a, \mathbf{x})$. In the context of transforming non-conformity scores, a natural additional requirement is that the ordering of the scores are retained, that is, $t_\theta$ should be strictly monotone in $a$.

There are many designs of CNFs, and for monotone transformations of one-dimensional scores we adopt conditional affine coupling layers (Dinh et al., 2016; Papamakarios et al., 2021). For a raw nonconformity score $a$ and features $\mathbf{x}$, the transformation to the new score $b$ is

$$b = a \cdot \exp\big(\sigma_\theta(\mathbf{x})\big) + \mu_\theta(\mathbf{x}), \tag{3}$$

where $\sigma_\theta(\cdot)$ and $\mu_\theta(\cdot)$ are neural networks that serve as flexible, feature-specific scale and shift terms. The exponential factor guarantees positivity, ensuring the mapping from $a$ to $b$ is strictly increasing, as required.

By the change-of-variables formula, the conditional density of $a$ given $\mathbf{x}$ is

$$p_{A|\mathbf{x}}(a \mid \mathbf{x}) = p_B\big(t_\theta(a, \mathbf{x})\big) \left| \tfrac{\partial}{\partial a} t_\theta(a, \mathbf{x}) \right|,$$

where $p_B$ is the chosen base density (e.g., standard normal or logistic) that is *free of* $\mathbf{x}$. Accordingly, the negative log-likelihood for a data point $(a_i, \mathbf{x}_i)$ is

$$\ell_\theta(a_i, \mathbf{x}_i) = -\log p_B\big(t_\theta(a_i, \mathbf{x}_i)\big) \ - \ \log \left| \tfrac{\partial}{\partial a} t_\theta(a_i, \mathbf{x}_i) \right|.$$

The parameters $\theta$ are estimated by minimizing the negative log-likelihood (NLL) of the transformed scores against the base density, which is

$$\mathcal{L}_{\text{NLL}}(\theta) = - \sum_{i \in \mathcal{D}_{\text{cal}}} \Big[ \log p_B\big(t_\theta(a_i, \mathbf{x}_i)\big) + \log \big|\partial_a t_\theta(a_i, \mathbf{x}_i)\big| \Big]. \tag{4}$$

A primary objective of finding $\theta$ that minimizes $\mathcal{L}_{\text{NLL}}$ is to align the conditional distribution $p_{B|\mathbf{X}=\mathbf{x}}$ with the common base density $p_B$ for all $\mathbf{x}$. The specific choice of $p_B$ is not crucial, as was confirmed in experiments that we conducted.

## 2.3 QUANTILE ALIGNMENT REGULARIZATION

While the flexibility of neural networks, $\sigma_\theta(\cdot)$ and $\mu_\theta(\cdot)$, enables CNFs to approximate the ideal transformation, their parameters $\theta$ are fitted using a finite sample of calibration data, and the independence of $B$ from $\mathbf{X}$ can not be perfectly achieved. In practice, residual dependence can persist and lead to coverage disparities across different regions of the feature space. To address this, we develop a *quantile alignment (QA)* regularizer.

Specifically, we partition the representation space into $G$ groups (e.g., via $k$-means clustering). Within each group $g$, we compute the empirical $(1 - \alpha)$-quantile of the transformed scores in the calibration set, denoted by $Q_g$, for $g = 1, \ldots, G$. If the conditional distribution of $B|\mathbf{X}$ were perfectly aligned across groups, all $Q_g$ would coincide. Deviations among these groupwise quantiles therefore serve as a measure of residual dependence on $\mathbf{X}$, and we aim to reduce it. Specifically, we define the QA penalty corresponding to the quantiles resulted from a given parameter value $\theta$ as

$$\mathcal{L}_{\text{QA}} \ = \ \frac{1}{G} \sum_{g=1}^{G} \big(Q_g - \overline{Q}\big)^2, \qquad \text{where } \overline{Q} = \tfrac{1}{G} \sum_{g=1}^{G} Q_g \, .$$

We modify the training of the CNF, $t_\theta$, from searching for $\hat\theta$ that minimizes the negative loglikelihood in equation 4 to that of the following target function:

$$\mathcal{L}_{\text{total}}(\theta) = \mathcal{L}_{\text{NLL}}(\theta) + \lambda \, \mathcal{L}_{\text{QA}}(\theta), \tag{5}$$

where $\lambda > 0$ is a tuning parameter that controls the trade-off between alignment with the base density and alignment among the subgroup quantiles. In practice, we found this modification to significantly improve stability and reliability of the prediction sets.

## 2.4 Determine thresholding Quantile via further calibration

After training $\hat{f}$ on $\mathcal{D}_{\text{train}}$ and $t_{\hat{\theta}}$ on a first calibration set, we need to obtain transformed scores $b$ and its empirical quantile $Q_{1-\alpha}$ on a second *held-out* calibration set $\mathcal{D}_{\text{cal}_2}$. That is, it was not involved in the training of $\hat{f}$ or $t_{\hat{\theta}}$. This is to ensure exchangeability of future data points and data in the held-out set to maintain validity of the coverage rate.

Finally, for a new input $\mathbf{x}_{n+1}$, we form the prediction set:

$$\widehat{C}_\alpha(\mathbf{x}) \;=\; \big\{\, y \in \mathcal{Y} : b(\mathbf{x}_{n+1}, y) \leq Q_{1-\alpha} \,\big\}. \tag{6}$$

A summary of the CONFLO algorithm is provided below. It outputs prediction sets with guaranteed marginal coverage of $(1 - \alpha)$ (Proposition 1). While the algorithm is new, the marginal coverage proof follows standard conformal prediction arguments; we state it as a proposition for clarity and citation. In addition, we have a conjecture that asymptotic conditional coverage holds for the CONFLO predictions under some regularity conditions. Briefly, as the total data size increases, if we split the data into training and calibration properly such that there is sufficient data for training with diminishing error both the predictor and the transformer within subpopulations of $\mathcal{X}$ of interest, then the transformed score $A$ becomes asymptotically independent of $\mathbf{X}$, and the resulting prediction sets achieve asymptotic conditional coverage.

Remark: in practice, a user can avoid empty prediction sets by opting to apply the standard "non-empty set" correction that adds a most likely value of $y$, $\arg\max_y \hat{p}(y \mid \mathbf{x})$. Clearly, this step will not reduce the coverage probability.

**Proposition 1.** *Suppose the sample points in calibration set and $(\mathbf{x}_{n+1}, y_{n+1})$ are exchangeable. For any prediction set obtain in equation 6, we have the following coverage guarantee:*

$$\mathbb{P}(y_{n+1} \in \widehat{C}_\alpha(\mathbf{x}_{n+1})) \geq 1 - \alpha.$$

*Furthermore, if the scores $B_i$ are almost surely distinct, the marginal coverage is near tight:*

$$\mathbb{P}(y_{n+1} \in \widehat{C}_\alpha(\mathbf{x}_{n+1})) \leq 1 - \alpha + \frac{1}{|\mathcal{D}_{\text{cal}}| + 1}.$$

---

**Algorithm 1** CONFLO

---

1: **Input:**
- Training data $\{(\mathbf{x}_i, y_i)\}_{i=1}^n \subset \mathbb{R}^p \times \mathbb{R}$
- Miscoverage level $\alpha \in [0, 1]$
- Specifications of the family of CNF, $\{t_\theta\}$, including the affine form, and neural networks $\mu$ and $\sigma$ in equation 3
- Test input $\mathbf{x}_{n+1}$ whose response $\mathbf{y}_{n+1}$ requires a prediction set

2: **Procedure:**
1. Randomly split the data into three disjoint subsets: $\mathcal{D}_{\text{train}}, \mathcal{D}_{\text{cal}_1}, \mathcal{D}_{\text{cal}_2}$.
2. Train a predictor or a classifier $\hat{f}$ on $\mathcal{D}_{\text{train}}$.
3. Form nonconformity scores $a$ (e.g. following equation 2) for subjects in $\mathcal{D}_{\text{cal}_1}$, and fit a CNF $t_{\hat{\theta}}$ by minimizing the target function in equation 5.
4. Obtain transformed score $b(\mathbf{x}_i, y_i) = \hat{t}(a(\hat{f}, \mathbf{x}_i, y_i), \mathbf{x}_i)$ for $i \in \mathcal{D}_{\text{cal}_2}$, and find the conformal threshold $Q_{1-\alpha}$.

3: **Output:** Prediction set for $y_{n+1}$ is given by

$$\widehat{C}_\alpha(\mathbf{x}_{n+1}) \;=\; \big\{\, y \in \mathcal{Y} : b(\mathbf{x}_{n+1}, y) \leq Q_{1-\alpha} \,\big\}.$$

---

**Dual-ascent tuning.** We optimize the weight $\lambda$ in equation 5 using a dual-ascent scheme. In this process, the normalizing flow is trained on $\mathcal{D}_{\text{cal}_1}$ for a total of $T$ epochs, divided into segments of length $t$. After each training segment, we re-evaluate empirical coverage on the same $\mathcal{D}_{\text{cal}_1}$ and

update $\lambda$ accordingly. If coverage falls below the target level $1 - \alpha$, $\lambda$ is increased to place more emphasis on the quantile alignment penalty; otherwise, it is decreased to preserve the likelihood-fitting objective. By iterating this train–evaluate–update cycle across all $T$ epochs, $\lambda$ is adaptively steered toward values that reduce inefficiency (e.g., set size or SSCV) while respecting the coverage constraint.

---

**Algorithm 2** Dual-ascent tuning for $\lambda$

---

1: **Input:** initial $\lambda_0$, steps $\eta_c, \eta_s > 0$, total epochs $E$, segment length $t$, training set $\mathcal{D}_{\mathrm{cal}_1}$, bounds $\lambda_{\min}, \lambda_{\max}$, scale $\beta > 0$
2: **Output:** tuned weight $\lambda^\star$
3: Initialize $\lambda \leftarrow \lambda_0$, $S \leftarrow \lceil E/t \rceil$
4: **for** $s = 1$ to $S$ **do**
5:      Train CNF on $\mathcal{D}_{\mathrm{cal}_1}$ for $t$ epochs with current $\lambda$
6:      Estimate coverage $\widehat{\mathrm{cov}}$ on $\mathcal{D}_{\mathrm{cal}_1}$
7:      Shortfall $g \leftarrow (1 - \alpha) - \widehat{\mathrm{cov}}$
8:      **if** $g > 0$ **then**                                             $\triangleright$ under-covered
9:          $\lambda \leftarrow \min\left\{\lambda_{\max}, \lambda + \eta_c \cdot g \cdot \beta\right\}$
10:     **else**                                                  $\triangleright$ coverage sufficient
11:          $\lambda \leftarrow \max\left\{\lambda_{\min}, (1 - \eta_s)\lambda\right\}$
12:     **end if**
13: **end for**
14: **return** $\lambda^\star \leftarrow \lambda$

---

## 3   RELATED WORK

We now review related work along two complementary lines: (i) adaptive set construction from classifier probabilities (APS, RAPS) and (ii) flow-based transformations that reshape outputs or residuals prior to calibration.

In APS (Romano et al., 2020), the nonconformity score for a pair $(x, y)$ is the (randomized) cumulative mass along the sorted softmax vector:

$$S_{\mathrm{APS}}(x, y, u; \hat{\pi}) \;:=\; \sum_{i=1}^{o(y, \hat{\pi}(x)) - 1} \hat{\pi}_{(i)}(x) \;+\; u\, \hat{\pi}_{(o(y, \hat{\pi}(x)))}(x), \qquad u \sim \mathrm{Unif}[0, 1],$$

where $o(y, \hat{\pi}(x))$ is the *rank* of $y$ among the probabilities $\hat{\pi}(x)$ sorted in decreasing order. A single threshold $\tau$ is then calibrated on a held-out split as the $(1 - \alpha)$ conformal quantile of these scores, yielding finite-sample *marginal* coverage $1 - \alpha$. At test time, the prediction set contains the top labels whose cumulative mass first exceeds $\tau$.

Because APS is sensitive to noisy probability estimates, the resulting sets are often larger than necessary. To address this, Angelopoulos et al. (2022) modified the APS score with a size penalty that discourages including many low-probability labels, together with a small top-$k_{\mathrm{reg}}$ buffer to stabilize the boundary. To optimize the penalty parameter $\lambda$ and evaluate conditional coverage, they further proposed the *size–stratified coverage violation* (SSCV) metric. Specifically, let $\{S_j\}_{j=1}^s$ be disjoint set–size strata such that $\bigcup_{j=1}^s S_j = \{1, \ldots, |\mathcal{Y}|\}$. The index set of examples whose prediction–set size (from algorithm $\mathcal{C}$) falls into stratum $S_j$ is

$$\mathcal{J}_j \;=\; \left\{\, i \in [n] : \left|\mathcal{C}(X_i, Y_i, U_i)\right| \in S_j \,\right\}.$$

Then SSCV is defined as

$$\mathrm{SSCV}(\mathcal{C}, \{S_j\}_{j=1}^s) \;=\; \sup_{j \in [s]} \left| \frac{\left|\left\{\, i : Y_i \in \mathcal{C}(X_i, Y_i, U_i), \, i \in \mathcal{J}_j \,\right\}\right|}{|\mathcal{J}_j|} \;-\; (1 - \alpha) \right|. \tag{7}$$

This metric measures the maximum deviation from the target coverage across different strata of prediction–set sizes.

However, APS and RAPS do not explicitly model or remove the $\mathbf{x}$-dependence of the nonconformity scores. They calibrate a single global threshold, guaranteeing only marginal coverage. As a result, subgroup- or $\mathbf{x}$-conditional reliability may improve but can still vary systematically.

This motivates flow-based conformal methods: first transform an $\mathbf{x}$-dependent quantity (scores, outputs, or residuals) into a simple base distribution through a bijective function, then calibrate in that latent space using one global quantile, and finally map back to obtain the prediction set or interval. In CONTRA (Fang et al., 2025), a flow sends the output to a latent variable that is approximately standard normal; nonconformity is then defined as the distance to the origin in latent space, so a single radius threshold becomes a compact prediction region after mapping back. Colombo (2024) proposed fitting a flow on residuals to reduce $\mathbf{x}$-dependence, after which a single quantile on the transformed residuals can be inverted to produce input-adaptive intervals.

## 4 EXPERIMENTS

In this section, we systematically compare the performance of the proposed CONFLO to APS and RAPS methods reviewed in section 3. We evaluate our approach on ten multi-class benchmarks spanning three classes: tabular, image, and text.

**(1) Tabular data.** *CoverType* (Blackard, 1998) and *SATImage* (Srinivasan, 1993) are modeled with a two-layer multilayer perceptron (MLP). Each MLP processes the input features through two fully connected layers with nonlinear activation.

**(2) Image data.** For *MNIST* (LECUN) and *Fashion-MNIST* (Xiao et al., 2017), we adopt a lightweight convolutional neural network (CNN) tailored for grayscale images, consisting of two convolutional layers followed by pooling and a fully connected classifier. For more complex datasets, *SVHN* (Netzer et al., 2011), *CIFAR-10* (Krizhevsky, 2009), and *CIFAR-100* (Krizhevsky, 2009), we use a compact CNN backbone with additional convolutional blocks and batch normalization to handle higher-resolution, color images and greater class diversity.

**(3) Text data.** *AG News* (Zhang et al., 2015), *20 Newsgroups* (Mitchell, 1997), and *Banking77* (Casanueva et al., 2020) are processed using a Transformer encoder (DistilBERT) (Sanh et al., 2019). Input sequences are tokenized and embedded, passed through the pre-trained DistilBERT layers, and then classified with a linear head attached to the [CLS] token representation (Devlin et al., 2019) .

**Data partitioning.** We construct disjoint splits for classifier training $\mathcal{D}_{\text{train}}$, calibration $\mathcal{D}_{\text{cal}}$, and testing $\mathcal{D}_{\text{test}}$ under two strategies:

1. **Tabular.** We partition the full dataset into 65% $\mathcal{D}_{\text{train}}$, 25% $\mathcal{D}_{\text{cal}}$, and 15% $\mathcal{D}_{\text{test}}$.
2. **Image and Text Data** We retain the *official* test set as $\mathcal{D}_{\text{test}}$ and partition the provided training set into 75% for $\mathcal{D}_{\text{train}}$ and 25% for $\mathcal{D}_{\text{cal}}$.

Table 1: Summary of dataset structures in experiments.

| Dataset | num. of classes | $n_1$ (training) | $n_2$ (calibration) | $n_3$ (test) |
|---|---|---|---|---|
| SATImage | 5 | 3857 | 1605 | 965 |
| CoverType | 7 | 12000 | 5000 | 3000 |
| MNIST | 10 | 45000 | 15000 | 10000 |
| FMNIST | 10 | 45000 | 15000 | 10000 |
| SVHN | 10 | 54842 | 18315 | 26032 |
| CIFAR 10 | 10 | 37500 | 12500 | 10000 |
| CIFAR 100 | 100 | 37500 | 12500 | 10000 |
| AG News | 4 | 15000 | 5000 | 3000 |
| 20 Newsgroup | 20 | 11307 | 4712 | 2827 |
| Banking 77 | 77 | 7502 | 2501 | 3080 |

For large datasets, we first draw a label stratified subsample and then apply the same policy to the subset, detailed in Table 1. When additional hyperparameter tuning or flow training is required (e.g.,

for RAPS or CONFLO), we further split the calibration set as

$$\mathcal{D}_{\text{cal}} = \mathcal{D}_{\text{cal}_1} \cup \mathcal{D}_{\text{cal}_2}, \qquad |\mathcal{D}_{\text{cal}_1}| = 0.3 \, |\mathcal{D}_{\text{cal}}|, \quad |\mathcal{D}_{\text{cal}_2}| = 0.7 \, |\mathcal{D}_{\text{cal}}|,$$

using $\mathcal{D}_{\text{cal}_1}$ for tuning/flow training and reserving $\mathcal{D}_{\text{cal}_2}$ exclusively for conformal calibration. Methods that do not require tuning (e.g., APS) calibrate on the full $\mathcal{D}_{\text{cal}}$. We then train a base classifier $g$ on $\mathcal{D}_{\text{train}}$. In CONFLO, after getting the base classifier $g$, we freeze its weights and define the raw nonconformity score as

$$a = -\log \hat{p}(y \mid \mathbf{x}).$$

The pair $(a, \mathbf{x})$ is then passed to the conditional flow model, and QA groups are formed by $k$-means clustering on $\mathbf{x}$ (default $G{=}20$ ), where the weight $\lambda$ is tuned via a coverage-driven dual-ascent controller.

Table 2: Empirical coverage, average set size, and SSCV for 90% conformal prediction sets across ten datasets. Results are averaged over 20 random splits with standard errors. The best (smallest) set size or SSCV in each row is shown in bold.

| Dataset | Method | Coverage ↑ | Set Size ↓ | SSCV ↓ |
|---|---|---|---|---|
| SATImage | APS | $0.956 \pm 0.008$ | $1.180 \pm 0.018$ | $0.063 \pm 0.009$ |
| | RAPS | $0.953 \pm 0.008$ | $1.161 \pm 0.020$ | $0.059 \pm 0.009$ |
| | CONFLO | $0.912 \pm 0.009$ | $\mathbf{1.013} \pm 0.007$ | $\mathbf{0.013} \pm 0.007$ |
| CoverType | APS | $0.942 \pm 0.003$ | $1.267 \pm 0.013$ | $0.080 \pm 0.001$ |
| | RAPS | $0.940 \pm 0.003$ | $1.254 \pm 0.014$ | $0.078 \pm 0.002$ |
| | CONFLO | $0.902 \pm 0.004$ | $\mathbf{1.082} \pm 0.008$ | $\mathbf{0.004} \pm 0.002$ |
| MNIST | APS | $0.994 \pm 0.001$ | $1.006 \pm 0.001$ | $0.094 \pm 0.001$ |
| | RAPS | $0.993 \pm 0.001$ | $1.006 \pm 0.001$ | $0.093 \pm 0.001$ |
| | CONFLO | $0.991 \pm 0.001$ | $\mathbf{1.000} \pm 0.001$ | $\mathbf{0.091} \pm 0.001$ |
| FMNIST | APS | $0.950 \pm 0.002$ | $1.070 \pm 0.006$ | $0.050 \pm 0.002$ |
| | RAPS | $0.947 \pm 0.002$ | $1.061 \pm 0.004$ | $0.047 \pm 0.002$ |
| | CONFLO | $0.939 \pm 0.003$ | $\mathbf{1.034} \pm 0.010$ | $\mathbf{0.039} \pm 0.003$ |
| SVHN | APS | $0.961 \pm 0.003$ | $1.101 \pm 0.016$ | $0.061 \pm 0.003$ |
| | RAPS | $0.958 \pm 0.003$ | $1.082 \pm 0.012$ | $0.058 \pm 0.003$ |
| | CONFLO | $0.939 \pm 0.005$ | $\mathbf{1.000} \pm 0.000$ | $\mathbf{0.039} \pm 0.004$ |
| CIFAR-10 | APS | $0.917 \pm 0.005$ | $1.555 \pm 0.063$ | $\mathbf{0.029} \pm 0.005$ |
| | RAPS | $0.915 \pm 0.005$ | $1.529 \pm 0.067$ | $0.032 \pm 0.005$ |
| | CONFLO | $0.899 \pm 0.006$ | $\mathbf{1.366} \pm 0.061$ | $0.038 \pm 0.008$ |
| CIFAR-100 | APS | $0.905 \pm 0.005$ | $11.233 \pm 0.671$ | $\mathbf{0.040} \pm 0.008$ |
| | RAPS | $0.904 \pm 0.005$ | $11.123 \pm 0.772$ | $0.041 \pm 0.013$ |
| | CONFLO | $0.899 \pm 0.004$ | $\mathbf{9.258} \pm 0.637$ | $0.081 \pm 0.005$ |
| AG News | APS | $0.951 \pm 0.006$ | $1.256 \pm 0.019$ | $0.058 \pm 0.011$ |
| | RAPS | $0.948 \pm 0.007$ | $1.236 \pm 0.017$ | $0.055 \pm 0.012$ |
| | CONFLO | $0.906 \pm 0.010$ | $\mathbf{1.024} \pm 0.009$ | $\mathbf{0.009} \pm 0.006$ |
| 20 Newsgroups | APS | $0.921 \pm 0.006$ | $4.032 \pm 0.082$ | $\mathbf{0.052} \pm 0.009$ |
| | RAPS | $0.920 \pm 0.006$ | $4.016 \pm 0.096$ | $0.061 \pm 0.023$ |
| | CONFLO | $0.903 \pm 0.009$ | $\mathbf{3.612} \pm 0.243$ | $0.053 \pm 0.023$ |
| Banking 77 | APS | $0.944 \pm 0.003$ | $4.373 \pm 0.141$ | $0.060 \pm 0.007$ |
| | RAPS | $0.940 \pm 0.005$ | $3.883 \pm 0.379$ | $0.062 \pm 0.005$ |
| | CONFLO | $0.903 \pm 0.008$ | $\mathbf{2.513} \pm 0.620$ | $\mathbf{0.024} \pm 0.017$ |

Table 2 reports the empirical coverage probability, average set size, and SSCV of the prediction sets. Coverage rates are generally close to the nominal 0.9 across all methods. On simpler datasets—such as the tabular benchmarks and the lower-complexity image datasets (MNIST, FMNIST, and SVHN)—all approaches produce very small sets, with CONFLO achieving not only the smallest set sizes but also the lowest SSCV. On the remaining five datasets, CONFLO continues

to yield the smallest prediction sets, often substantially smaller than the alternatives. In terms of SSCV, CONFLO achieves the lowest values on two benchmarks while remaining broadly comparable on the others. Overall, CONFLO provides a single-threshold approach that reliably meets the target marginal coverage while producing compact prediction sets, and further improves conditional coverage reliability through appropriate tuning of the QA strength or group formation.

## 5 SUMMARY AND DISCUSSION

Our approach, CONFLO[2], builds on the central idea of flow-based transformations for conformal prediction, but introduces two key innovations that substantially broaden applicability and improve conditional reliability.

First, unlike prior methods that apply normalizing flows to residuals or outputs, CONFLO is applicable to outcomes of any type—whether continuous, multidimensional, or categorical. This flexibility is achieved by applying a conditional normalizing flow (CNF) directly to a user-chosen nonconformity score, rather than to the joint distribution of residuals and covariates. By decoupling the framework from the outcome type, CONFLO provides a unified tool for conformal prediction across diverse problem settings.

Second, instead of relying solely on the transformation to induce conditional independence, CONFLO explicitly enforces groupwise quantile alignment (QA) through an additional penalty term in the CNF training objective. This regularization ensures that the transformed scores exhibit near-invariance across subgroups of the input space, thereby approximate the conditional coverage.

Although CONFLO shows strong empirical performance across diverse tasks, our results so far provide only empirical evidence of conditional reliability. A key limitation is that we have not yet delivered a full theoretical proof of its asymptotic near-conditional coverage guarantees. In future work, we plan to formalize these guarantees with rigorous asymptotic analysis, providing a solid theoretical foundation to complement our empirical findings.

---

[2]Use of Large Language Models: LLMs were used to polish the writing and to help search potentially relevant methods and related work. All technical contributions, analyses, and conclusions are solely those of the authors.

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
