# OpenReview forum: "CONFLO: Conformal Prediction with Conditional Coverage via Normalizing Flow"
_ICLR.cc/2026/Conference — Submitted to ICLR 2026_

### Official Review · Reviewer_haGN · 2025-10-27

**Soundness:** 3
**Presentation:** 2
**Contribution:** 3
**Rating:** 2
**Confidence:** 2

**Summary:**

This paper introduces CONFLO to achieve improved conditional coverage across subpopulations by integrating Conditional Normalizing Flows (CNF) with a quantile alignment (QA) regularization strategy.
CONFLO consists of two key steps: Score Transformation and Quantile Alignment Regularization.
CONFLO also introduces an adaptive dual-ascent tuning to automatically balance likelihood fitting and quantile alignment, improving stability and coverage control.

**Strengths:**

1. Applying conditional normalizing flows into CP is interesting. By transforming nonconformity scores rather than residuals or outputs, CONFLO generalizes beyond regression to both categorical and multidimensional outcomes.

2. CONFLO unifies multiple previous ideas, such as multicalibration, flow-based reparametrization, and conformal calibration, to a framework.

3. The training process (the algorithmic presentation in Algorithms 1 and 2) is clear.

**Weaknesses:**

1. Although the paper conjectures asymptotic conditional coverage under certain regularity conditions, it does not provide any formal proof or theoretical bounds. Given the paper’s title and motivation, the lack of further theoretical discussion weakens the rigorness, especially for an ICLR main-track submission.

2. Can you formalize or empirically demonstrate the conditions under which the transformed score $B = t_\theta(A, X)$ becomes asymptotically independent of X?

3. The method involves training CNFs and computing groupwise quantiles at every iteration, which introduces extra computation compared to standard CP methods. The paper lacks emprical and theoretical analysis of computational cost (comparison to APS/RAPS or other baselines).

4. The quantile alignment regularizer depends on k-means and $\lambda$, both of which may introduce sensitivity. However, there is no ablation study or sensitivity analysis exploring how they affect coverage and efficiency. The effects of (a) CNF transformation alone vs. (b) CNF + QA regularization are also missed. Besides, does $\lambda$ converge to a stable value across runs?


5. CONFLO behind baselines on CIFAR-100 and 20 Newsgroups, whose number of classes is larger than other datasets. Its SSCV is wrose than RAPS. Does it means that the CONFLO’s benefits may diminish as data complexity increases?

6. The baselines and related work miss more recent CP methods, where the only baselines are APS and RAPS.

7. Concerning writing and presentation:

7a. The big picture (how the **subpopulation** issue is addressed by **CONFLO**) is not clear.

7b. Some sections (especially 2.2–2.3) are dense and could be better motivated with intuition or figures illustrating the effect of CNF transformations and quantile alignment.

7c. Key information in Figure 1 is hard to distinguish, such as legned and texts.

**Questions:**

See above.

---

### Official Review · Reviewer_54DC · 2025-10-30

**Soundness:** 2
**Presentation:** 2
**Contribution:** 1
**Rating:** 4
**Confidence:** 4

**Summary:**

This paper proposes CONFLO, a conformal prediction framework that aims to achieve conditional coverage by using conditional normalizing flows (CNF) to transform non-conformity scores into new scores that are nearly independent of input features.

**Strengths:**

Systematic evaluation across ten diverse datasets spanning tabular (CoverType, SATImage), image (MNIST, CIFAR-10/100, SVHN), and text domains (AG News, Banking77).

**Weaknesses:**

Marginal coverage guarantee (Proposition 1) is standard and doesn't address the main conditional coverage claim.

No theoretical analysis of when/why the QA regularization should work or under what conditions near-independence can be achieved.

Builds incrementally on existing flow-based conformal methods. It is unclear how their CNF application to scores differs from existing works.

Only compares against APS and RAPS; missing comparisons to other conditional conformal prediction variants.

APS and RAPS overcover in many datasets, indicating potential implementation issues or unfair baseline setup.

QA regularization appears ad-hoc without strong theoretical justification for why minimizing quantile differences should improve conditional coverage. The number of groups is not justified or validated.

Computational overhead of training CNF not discussed or compared against baselines.

Hyperparameter sensitivity (number of groups G, clustering method) not thoroughly analyzed.

**Questions:**

The authors should address the concerns raised in the weakness section above.

---

### Official Review · Reviewer_Gnzi · 2025-11-01

**Soundness:** 2
**Presentation:** 3
**Contribution:** 1
**Rating:** 2
**Confidence:** 4

**Summary:**

This paper proposes to address the conditional validity of conformal prediction by using conditional normalizing flows to transform the scores, such that the transformed conformity score is independent of the feature. There are two novel aspects compared to Colombo et al. First, CONFLO is applicable to any outcome structure including continuous and discrete outcomes. This is achieved by transforming non-conformity scores instead of raw residuals. Second, the conditional normalizing flow is trained with a regularizer to explicitly enforce quantile alignment of transformed scores across feature groups. The resulting method is validated on tabular, image and language datasets, which show improved conditional coverage and reduces set size compared to APS and RAPS.

**Strengths:**

1. This paper studies the core challenge for the conditional validity of conformal prediction, which limits the application of conformal prediction in OOD cases.
2. The proposed method has demonstrated significant improvement in conditional validity in 6 of 10 tasks, compared to APS and RAPS.

**Weaknesses:**

1. Regression tasks are not involved in experiments, though the proposed method is theoretically compatible with regression tasks.

2. The method is benchmarked against basic conformal prediction algorithms such as APS, which does not guarantee conditional validity. There is a branch of research that provides conditional validity guarantee [1,2,3]. Some of these methods are discussed as related work but none of them are shown in experiments.

3. It is unclear what technical challenge arises considering transformation of conformity scores instead of residuals.

4. It is unclear what degree of improvement can be attributed to the quantile alignment regularizer. An ablation study will be relevant.

5. A conditional coverage guarantee is not theoretically provided, in contrast to related research [1,2,3].

[1] Isaac Gibbs, John J. Cherian, and Emmanuel J. Cand`es. Conformal prediction with conditional guarantees, 2024. URL https://arxiv.org/abs/2305.12616.
[2] Jung, C., Noarov, G., Ramalingam, R., & Roth, A. (2022). Batch multivalid conformal prediction. arXiv preprint arXiv:2209.15145.
[3] Bairaktari, K., Wu, J., & Wu, Z. S. (2025). Kandinsky Conformal Prediction: Beyond Class-and Covariate-Conditional Coverage. arXiv preprint arXiv:2502.17264.

**Questions:**

Please refer to the weaknesses.

---

### Official Review · Reviewer_yVMa · 2025-11-01

**Soundness:** 2
**Presentation:** 2
**Contribution:** 2
**Rating:** 4
**Confidence:** 4

**Summary:**

The paper proposes a novel conformal prediction method based on conditional normalized flows, CONFLO. This method aims to obtain smaller prediction set sizes and better conditional coverage than standard methods, by virtue of the normalizing flow insight. Thereby, the set of original nonconformity scores are mapped via a special trainable set of mappings to “approximately feature-independent” nonconformity scores; the principle is that on the transformed scores, it is possible to use the marginal quantile for the cutoff directly (as in vanilla marginal conformal prediction), instead of applying adaptive conformal methods, while still ensuring adaptively changing prediction sets as a result of the inverse flow map. In addition, the further quantile correction component is added to the algorithm, which regularizes by enforcing the rough equality of group quantiles (as should happen under true independence) on some heuristically derived groups. Empirical results demonstrate improvements over APS and RAPS in terms of vanilla coverage, stratified conditional coverage, and prediction set size, on a suite of experimental tasks.

**Strengths:**

For the strengths of the paper, several aspects stand out. First of all, the improved performance in terms of adaptivity (prediction set size and stratified conditional coverage), while keeping the tight marginal coverage guarantees, are important.

Second, the new angle on the use of normalizing flows in conformal prediction (namely, applying them directly on the nonconformity scores) is quite clean and useful, and leads to a broader applicability of the method (as it can thus be used both for regression and for classification problems without modification).

Third, the quantile regularization technique is a quick but useful idea; in a sense, it brings back the group-wise quantile conditions idea in this simplified, CNF-adjusted form.

**Weaknesses:**

First, the paper does a bit of a restricted job in terms of conditional coverage guarantees. On the theoretical front, we only have a conjecture of asymptotically conditional coverage, which appears plausible from a general perspective but the (likely slow, as suggested by results concerning the impossibility of exact conditional coverage) quantitative rates would be key to have.

But more immediately, in the context of the manuscript, even the empirical evaluation of conditional coverage is not fully convincing, For SSCV, the strata must be selected heuristically, and in this case 20 such strata were created — but no ablation was performed against other granularity or other strata construction methods (such as actual meaningful, possibly intersecting, groups); as such, the obtained empirical conditional coverage rates are suggestive but not conclusive or thorough enough.

Further, for the current results, while SSCV was in fact much better for CONFLOW on several settings, but on other settings it was quite close of a gap, so given the “simple baseline” status of APS and RAPS, the results may not be very impressive just yet.

Second, developing the above point, there are only comparisons to APS/RAPS nonconformity scores. As such, it’s not clear to me why the method was not compared to conditional coverage conformal methods. And, importantly, why no comparison was made to other normalized flow-based conformal methods; these methods were discussed in the intro and related work sections, but very briefly — this needs to be addressed, both in terms of a longer qualitative discussion of these other 2 methods, and in terms of finding a way to include them as baselines.

Furthermore, the text doesn’t elaborate on any ablation with respect to the quantile-fixing regularization procedure. Indeed, it’s not clear whether the weight is pulled mainly by the CNF, or if the main role of the CNF is to transform the nonconformity scores such that the quantile equating regularization term can become meaningful and pull the weight? As such, I believe empirical evaluation on this ablation front is needed.

**Questions:**

Please see above in the weaknesses section; many of my questions concern conditional coverage, a better comparison to related work, and more careful ablations.

---

### Meta-Review · Area_Chair_anVg · 2026-01-09

**Summary:**

This paper considers the problem of conditional validity within the conformal prediction framework for uncertainty quantification. It develops an approach using conditional normalizing flows where the key idea is to transform non-conformity scores into new scores that are nearly independent of input features.

The reviewers have identified a number of weaknesses of this paper including:
- Lack of rigorous theoretical analysis for conditional coverage of the approach
- Lack of experimental comparison with baseline CP methods for conditional coverage
- Paper could benefit from stronger motivation for transformation of scores and additional ablation analysis

Therefore, I recommend to reject this paper and encourage the authors' to address them for re-submission to a future venue.

**Reviewer Concerns:**

N/A. No rebuttal was submitted.

**Reviewer Scores:**

N/A

---

### Decision · Program_Chairs · 2026-01-26

Reject